# Gut–Liver Axis-Mediated Anti-Obesity Effects and Viscosity Characterization of a Homogenized Viscous Vegetable Mixture in Mice Fed a High-Fat Diet

**DOI:** 10.3390/plants14162510

**Published:** 2025-08-12

**Authors:** Yu-An Wei, Yi-Hsiu Chen, Lu-Chi Fu, Chiu-Li Yeh, Shyh-Hsiang Lin, Yuh-Ting Huang, Yasuo Watanabe, Suh-Ching Yang

**Affiliations:** 1School of Nutrition and Health Sciences, Taipei Medical University, Taipei 11031, Taiwan; ba06110029@tmu.edu.tw (Y.-A.W.); da07113003@tmu.edu.tw (Y.-H.C.); ma07111003@tmu.edu.tw (L.-C.F.); clyeh@tmu.edu.tw (C.-L.Y.); lin5611@tmu.edu.tw (S.-H.L.); da07110003@tmu.edu.tw (Y.-T.H.); 2Research Center of Geriatric Nutrition, College of Nutrition, Taipei Medical University, Taipei 11031, Taiwan; 3School of Food Safety, Taipei Medical University, Taipei 11031, Taiwan; 4Medgaea Life Science Ltd., New Taipei City 351007, Taiwan; 5General Health Medical Center, Yokohama University of Pharmacy, Kanagawa 245-0066, Japan; yasuwat@yok.hamayaku.ac.jp; 6Nutrition Research Center, Taipei Medical University Hospital, Taipei 11031, Taiwan; 7School of Gerontology and Long-Term Care, College of Nursing, Taipei Medical University, Taipei 11031, Taiwan

**Keywords:** anti-obesity, viscous vegetable mixture, non-alcoholic fatty liver disease, gut microbiota, mice

## Abstract

This study investigated the anti-obesity effects of a homogenized, viscous vegetable (VV) mixture prepared from mucilaginous vegetables, with a focus on modulating hepatic lipid metabolism and gut microbiota composition in mice fed with a high-fat (HF) diet. The VV mixture was formulated by blending freeze-dried powders of ten mucilaginous vegetables, classified as moderately thick using a line-spread test and extremely thick according to the IDDSI framework in a 1:9 ratio (VV mixture: water, *w*/*w*). Six-week-old male C57BL/6 mice were fed control or HF diets, with or without 10% VV mixture for 8 weeks (*n* = 7 per group). The HF diet induced significant weight gain, adipose tissue accumulation, hepatic steatosis, and inflammation. The HF diet also significantly reduced hepatic ACO1, CPT1 mRNA expression, and α-diversity with distinct fecal microbiota profiles. On the other hand, VV mixture supplementation reduced serum TC, LDL-C levels and NAFLD scores. VV mixture supplementation also increased hepatic ACO1 and CPT1 mRNA expression, enhanced α-diversity, and enriched SCFA-producing bacteria, particularly the *Lachnospiraceae NK4A136* group. In conclusion, the VV mixture attenuated HF diet-induced obesity, possibly through its high viscosity–mediated effects on hepatic fatty acid oxidation and gut microbiota modulation.

## 1. Introduction

Obesity is a growing global health concern associated with metabolic disorders, liver dysfunction, and alterations in the gut microbiota [1]. The World Health Organization (WHO) reported that over 2.5 billion adults were overweight and obese (with a body-mass index (BMI) of ≥25 kg/m^2^) in 2020, among which 890 million were living with obesity [2]. Moreover, metabolic-associated fatty liver disease (MAFLD), recently renamed from non-alcoholic fatty liver disease (NAFLD) to highlight the central role of metabolic dysfunction. It is now recognized as the most common chronic liver disease worldwide, frequently coexisting with obesity, dyslipidemia, insulin resistance, and other metabolic disorders [3]. In addition to metabolic dysfunction, MAFLD has been linked to gut microbiota dysbiosis triggered by unhealthy dietary habits [4,5]. A high-fat (HF) diet has emerged as one of the most potent triggers impairing gut epithelial barrier function, resulting in elevation of intestinal permeability and endotoxemia, thereby mediating liver inflammation [6]. In terms of the gut microbiotic composition, an HF diet is associated with an increased abundance of the Bacteroides enterotype and an elevated Firmicutes/Bacteroidetes (F/B) ratio [7]. Taken together, obesity, MAFLD, and gut microbiota dysbiosis are closely interconnected, forming a complex pathological network that reinforces disease progression and highlights the need for integrative dietary strategies targeting the gut-liver axis. Although no pharmacological treatment for NAFLD has been approved, the primary approach remains lifestyle modification, particularly through diet modification and exercise [8]. Research evidence suggests that reducing caloric intake through high dietary fiber consumption can promote weight loss and improve insulin resistance, hepatic steatosis, and even fibrosis [9], as dietary fiber contributes to weight reduction by promoting satiety and a low energy density [10].

Mucilage is a sticky, mucus-like substance present in certain vegetables with viscous characteristics. The main constituents of mucilage include polysaccharides, proteins, minerals, lipids, and uronic acid units, which contribute to the unique texture of these vegetables [11]. Mucilage, which contains polysaccharides that serve as precursors for short-chain fatty acids (SCFAs), is associated with several health benefits, including laxative, hypolipidemic, antihyperglycemic, antioxidative, and antibacterial effects [12,13]. SCFAs play a critical role in maintaining the intestinal barrier integrity by regulating luminal pH, stimulating mucus secretion, and providing energy to epithelial cells [14]. Additionally, polysaccharides were shown to increase the microbial diversity and abundances of beneficial bacteria such as *Lactobacillus* and *Bifidobacterium*. Furthermore, polysaccharides influence lipid metabolism by enhancing the Farnesoid X receptor (FXR)-related signaling pathway and upregulating cholesterol 7-alpha-hydroxylase (CYP7A1) activity, which leads to inhibition of hepatic lipogenesis and promotes the conversion of cholesterol [15]. Recent studies focused on polysaccharide extracts from individual mucilaginous vegetables such as seaweed (Laminariaceae), kelp (*Undaria pinnatifida*), *Tremella fuciformis*, *Flammulina velutipes*, and okra (*Abelmoschus esculentus*) [16,17,18,19]. Further, a few studies investigated the effects of whole, unprocessed, and unpurified blends of these vegetables as combined dietary interventions [20,21,22,23].

While most studies focused on purified polysaccharides from individual mucilaginous vegetables, few have explored whole-food blends that better reflect real-life diets. Such combinations may offer broader nutritional benefits and enhanced viscosity, making them more suitable for people in need of texture-modified or thickened diets, such as older adults. In this study, we investigated the metabolic effects of a homogenized mucilaginous vegetable mixture in mice fed an HF diet, with a focus on the gut-liver axis, while also characterizing the viscosity of the mixture to assess its potential suitability for texture-modified diets.

## 2. Results

### 2.1. Nutrient Composition and Viscosity Analysis

Nutrition components of the viscous vegetable (VV) mixture are shown in Table 1 and Table 2. Okra had the highest polysaccharide and polyphenol contents. Based on these results, daily supplementation with 350 mg of the VV mixture provided approximately 74.2 mg of polysaccharides and 4.2 mg gallic acid equivalents (GAE) of polyphenols (Table 2).

At a 1:9 ratio of mucilaginous ingredients to water, the line spread test (LST) grading method classified the VV mixture as moderately thick, while the International Dysphagia Diet Standardisation Initiative (IDDSI) framework classified the VV mixture as extremely thick.

### 2.2. Obesity-Related Indicators

#### 2.2.1. Food Intake and Body Weight (BW) Gain

Compared to the C group, the CV and H groups showed significantly lower daily energy intake levels (Figure 1A). However, no significant differences in daily energy intake were observed among the H and HV groups (Figure 1A). As shown in Figure 1B, the CV group showed a significantly lower BW gain, whereas the H group exhibited a significantly greater BW gain compared to the C group. Notably, the HV group also showed a significantly reduced BW gain relative to the H group (Figure 1B). In addition, the H group demonstrated a higher food efficiency ratio (FER) compared to the C group, whereas the HV group exhibited a lower FER than the H group (Figure 1C). When compared to the CV group, only the HV group showed a higher FER (Figure 1C).

#### 2.2.2. Adipose Tissue Weights

The H group showed significantly higher relative weights of perirenal, mesenteric, and epididymal white adipose tissues (WATs) compared to the C group (Figure 1D–F). However, the HV group exhibited significantly less mesenteric and epididymal WAT weights than the H group (Figure 1E,F). Moreover, compared to CV group, the HV group presented significantly higher perirenal and epididymal WAT weights (Figure 1D,F).

#### 2.2.3. Adipocyte Size

Histological results of epididymal adipose tissues are shown in Figure 1G. Compared to the C group, adipocytes were markedly enlarged in the H group. However, the adipocyte size was visibly smaller in the HV group than that in the H group (Figure 1G). A quantitative analysis revealed that the mean adipocyte size was significantly larger in the H group than in the C group, whereas the HV group showed a significant reduction in adipocyte size compared to the H group (Figure 1H).

### 2.3. Hepatic Damage

#### 2.3.1. Liver Function Index

No significant change was observed in serum aspartate aminotransferase (AST) activity in any groups (Figure 2A). In contrast, serum alanine aminotransferase (ALT) activity was significantly elevated in the H group compared to the C group but was markedly reduced in the HV group compared to the H group (Figure 2B).

#### 2.3.2. Histopathological Examinations and Lipid Peroxidation

Based on histological observations, liver steatosis and inflammatory cell infiltration were evident in the H group, and these pathological changes were significantly ameliorated by the SF intervention (Figure 2C). Compared to the C group, the H group showed a tendency toward increased NAFLD scores, whereas the HV group exhibited a significant reduction in NAFLD scores (Figure 2D). Moreover, no significant differences in lipid peroxidation levels were observed among the groups (Figure 2E).

### 2.4. Lipid Metabolism-Related Factors

#### 2.4.1. Serum Lipid Profiles

Compared to the C group, serum triglyceride (TG) levels were significantly decreased in both the CV and H groups (Figure 3A). There were no significant differences in serum total cholesterol (TC), low-density lipoprotein cholesterol (LDL-C), high-density lipoprotein cholesterol (HDL-C) concentrations among the C, CV, and H groups (Figure 3B–D). However, the HV group exhibited significantly lower TC and LDL-C serum levels compared to the H group (Figure 3B,C). In addition, compared to the CV group, the HV group showed a reduction in the serum LDL-C level and LDL-C/HDL-C ratio (Figure 3C,E).

#### 2.4.2. Hepatic TC and TG Concentrations

As shown in Figure 3F, no change was found in the hepatic TG level among all groups, while hepatic TC concentrations significantly decreased in the CV and H groups compared to the C group (Figure 3G). Furthermore, The HV group displayed a slightly lower liver TC level compared to the H group (Figure 3G).

#### 2.4.3. Plasma Adipokine Levels

Compared to the CV group, the HV group showed a significant decrease in plasma adiponectin levels and a significant increase in plasma leptin levels (Figure 3H,I). Compared to the C group, the H group exhibited a significantly higher plasma leptin level (Figure 3I), whereas the HV group showed a significant reduction in plasma leptin levels compared to the H group (Figure 3I). Additionally, the HV group demonstrated a significantly higher adiponectin/leptin ratio than the H group (Figure 3J).

#### 2.4.4. Lipid Metabolism-Related mRNA Levels

There were no significant changes in mRNA expressions related to fatty acid synthesis among all groups (Figure 4A–E). However, regarding mRNA expressions of fatty acid oxidation-related genes, the H group showed a significant reduction in acyl-CoA oxidase 1 (ACO1) and carnitine palmitoyltransferase 1 (CPT1) compared to the C group, both of which were restored in the HV group (Figure 4L,M). In terms of cholesterol metabolism-related genes, expression levels of sterol regulatory element-binding protein 2 (SREBP2) and cholesterol 7α-hydroxylase (CYP7A1) showed no significant differences among the groups (Figure 4N,O).

### 2.5. Intestinal Damage

#### 2.5.1. Relative Intestine Length

Compared to the C group, the CV group exhibited a significantly longer small intestine (Figure 5A). In contrast, both small intestine and colon lengths were significantly reduced in the H group relative to the C group (Figure 5A,B). Notably, the HV group showed a reversal of these effects, with a significantly increased small intestine length compared to the H group (Figure 5A).

#### 2.5.2. Fecal Microbiotic Analysis

An increased proportion of the Firmicutes was seen in the H group compared to the C group (Figure 5C). However, compared to the CV group, the HV group showed a significant increase in the proportion of the Firmicutes and a significant decrease in the proportion of the Bacteroidota, resulting in a higher Firmicutes/Bacteroidota (F/B) ratio (Figure 5C–E). For the α-diversity analysis, Chao1 was significantly decreased in the H group but significantly improved in the HV group (Figure 5F). The HV group also presented a significantly higher Shannon and Simpson index compared to the H group (Figure 5F). To assess variations in the fecal microbiota, a β-diversity analysis was performed using a PCoA plot. As shown in Figure 5G, compared to the C group, the H group exhibited a distinct gut microbiotic composition, whereas the CV and HV groups showed microbial profiles more similar to that of the C group.

The LEfSe approach and linear discriminant analysis (LDA) scores were used to identify bacterial taxa with significantly different abundances among the experimental groups. In the C group, the Rikenellaceae (family) and *Alistipes* (genus) of the Bacteroidota phylum were identified as dominant taxa (Figure 6B). In the CV group, the dominant bacterial taxa included the Bacteroidota (phylum), Bacteroidia (class), and Muribaculaceae (family) of the Bacteroidota phylum, as well as Acholeplasmatales (order) of the Mycoplasmatota phylum (Figure 6B). In the H group, enriched taxa included Actinobacteriota and Proteobacteria (phyla), and Enterobacterales (order) of the Proteobacteria phylum, Bacteroidaceae and Tannerellaceae (families) of the Bacteroidota phylum, and Ruminococcaceae (family), and *Negativibacillus*, *Roseburia*, and *Tuzzerella* (genera) of the Firmicutes phylum (Figure 6B). In the HV group, dominant taxa included Clostridia (class), Oscillospirales and Lachnospirales (orders), Butyricicoccaceae and *Lachnospiraceae_NK4A136_group* (family/genus), and *Acetatifactor* (genus), all of which belong to the Firmicutes phylum (Figure 6B).

## 3. Discussion

### 3.1. Anti-Obesity Effects of the Homogenized VV Mixture

#### 3.1.1. Viscosity Analysis

Commonly used food texture classification systems include IDDSI levels [24] and the Universal Design Food (UDF) classification established by the Japan Care Food Conference [25]. Moreover, it was demonstrated that the LST is a quick, objective, and visually interpretable method, potentially useful for helping patients and caregivers achieve more accurate and consistent preparation of thickened liquids [26]. In this study, the VV mixture diluted at a 1:9 ratio was classified as “moderately thick” and “extremely thick”, respectively, using the LST and IDDSI systems, indicating a discrepancy between the two methods (Table 3 and Table 4). In the future, the classification can be further verified using UDF standards by measuring quantitative viscosity data with a viscometer. According to UDF 2007 guidelines, food categorized under UDF level 3 typically has a viscosity ranging from approximately 150 to 1500 mPa·s, while level 4 foods exceed 1500 mPa·s [27]. In this study, the VV mixture demonstrated potential as a natural thickening agent, suitable for increasing food viscosity and serving as a nutrient-rich thickener composed of naturally derived, complex water-soluble fibers for individuals with swallowing difficulties [28].

#### 3.1.2. BW, and Body Fat and Calorie Intake

In the present study, although the HF diet did not increase overall energy intake, it led to significant weight gain and increased adiposity (Figure 1A,B). Supplementation with the VV mixture effectively reduced both BW and fat accumulation. In addition, the VV mixture also decreased the adipocyte size (Figure 1G,H). Under a normal diet, supplementation with the VV mixture also led to a reduction in mesenteric and epididymal WATs (Figure 1E,F). Dietary fiber is associated with a reduction in body fat through mechanisms involving enhanced satiety, decreased energy intake, and an improved gut microbiotic composition [29,30]. Several studies demonstrated that high-viscosity dietary fibers, including guar gum, β-glucans, mucilage, and alginates, can attenuate BW gain in mice fed an HF diet [18,21,22,31,32]. Tremella-derived polysaccharides were also shown to have beneficial effects in preventing adiposity [33]. Moreover, a recent meta-analysis further supported that seaweed supplementation, particularly in refined or extract forms, exerted positive effects on the body-mass index (BMI) and fat mass [34]. Collectively, those findings support the anti-obesity potential of polysaccharide-rich functional foods through modulation of BW and fat accumulation.

#### 3.1.3. Daily Dosage

In the present study, the average daily feed intake of the CV and HV groups ranged 2.7–3.2 g, of which 0.27–0.32 g was composed of the VV mixture. Based on its dietary fiber content (42.1 g/100 g), the estimated daily fiber intake was approximately 0.11–0.13 g per mouse, equivalent to 3.4–4.1 g/kg BW. When converted to a human equivalent dose for a 60 kg adult using a metabolic body surface area factor of 0.081, this corresponds to a daily fiber intake of approximately 16.5–19.9 g [35]. The recommended total dietary fiber is 25 g/day from WHO and non-starch polysaccharide (NSP) 18 g/day from the Committee on Medical Aspects of Food and Nutrition Policy [36,37]. The converted dosage in this study was close to the recommended NSP intake, which can be linked to higher fiber intake, reduced blood lipid, liver lipid accumulation, and anti-obesity effects [38].

### 3.2. Hepatic Protective Effects of the Homogenized VV Mixture

In this study, the HF diet induced an elevation in serum ALT activity, accompanied by a trend toward an increased NAFLD score (Figure 2B–D). However, supplementation with the VV mixture significantly reduced ALT activity and decreased the NAFLD score, indicating improvements in hepatic steatosis and inflammation (Figure 2B–D). Previous studies reported that kelp (*Undaria pinnatifida*) supplementation reduced adipocyte sizes and hepatic lipid accumulation in HF diet-fed C57BL/6J mice [21,36]. Similarly, white tremella (*Tremella fuciformis*) polysaccharides and okra-derived preparations were shown to attenuate hepatic lipid accumulation, including whole fruit powder, crude polysaccharides, and complex extracts [18,22,23,31,33,39]. Those findings suggest that viscous polysaccharides may modulate lipid metabolism by influencing pathways involved in lipogenesis and/or lipolysis.

### 3.3. Lipid Metabolism Regulation by the Homogenized VV Mixture

#### 3.3.1. Serum and Hepatic Lipid Profiles

The VV mixture significantly reduced serum TG and hepatic TC levels in mice fed a normal diet (Figure 3A,G, C vs. CV groups). This hypolipidemic effect may be attributed to its rich content of viscous polysaccharides and polyphenols, which are known to modulate lipid absorption, inhibit hepatic lipogenesis, and enhance fatty acid oxidation [33]. Notably, individual components such as Tremella polysaccharides and okra extracts were previously reported to exert TG-lowering effects [18,31]. These findings suggest that the VV mixture has potential as a functional dietary intervention for improving lipid metabolism even under normal physiological conditions.

Unexpectedly, a decrease in serum TG levels was observed in the H group (Figure 3A). This paradoxical effect was reported in previous studies and may be explained by several mechanisms. First, HF feeding can induce hepatic insulin resistance, which impairs very-low-density lipoprotein (VLDL) secretion and leads to TG accumulation in the liver rather than its release into the circulation [40,41]. Second, increased activity of lipoprotein lipase in peripheral tissues such as adipose tissues may enhance the hydrolysis and uptake of circulating TGs, resulting in reduced serum TG levels [42]. Third, there was a high portion of refined carbohydrate in the control group diet, which can cause the increased blood TG level [43]. In this study, although hepatic TG levels did not increase (Figure 3F), the histological analysis clearly revealed lipid accumulation in the H group, suggesting the presence of steatosis (Figure 2C). Future studies should investigate VLDL secretion to clarify this discrepancy.

The HF diet did not significantly affect any of the serum cholesterol parameters compared to the C group (Figure 3B–D). However, supplementing the VV mixture into the HF diet (HV group) resulted in a significant reduction in serum TC and LDL-C levels C (Figure 3B–D). Moreover, compared to the CV group, significant decreases in plasma LDL-C levels and the LDL-C/HDL-C ratio were observed in the HV group (Figure 3B–D). This suggests that the cholesterol-lowering effect was not merely a reversal of HF diet-induced dyslipidemia, but rather an independent effect of the VV mixture. The hypocholesterolemic effect may be attributed to the polysaccharide- and polyphenol-rich profile of the VV mixture, which was shown to inhibit cholesterol absorption, modulate bile acid metabolism, and enhance hepatic cholesterol catabolism via CYP7A1-mediated pathways [18,44]. Furthermore, dietary fibers with high viscosity were reported to bind bile acids in the intestines and promote their excretion, thereby stimulating hepatic conversion of cholesterol into bile acids [30]. Given the absence of significant changes in cholesterol metabolism-related genes such as *SREBP2* and *CYP7A1* (Figure 4N,O), future studies should focus on elucidating the regulatory mechanisms underlying bile acid secretion and metabolism.

In this study, it was found that an HF diet induced lower hepatic TC levels (Figure 3G, C vs. H groups). Although an HF diet typically leads to hepatic lipid accumulation, the observed decrease in hepatic TC levels may reflect a compensatory increase in cholesterol efflux or bile acid synthesis [45,46]. Additionally, dietary components or shifts in lipid storage preferences toward TGs may also have contributed to this unexpected reduction [47].

#### 3.3.2. Hepatic Lipid Metabolism-Related Factors

An HF diet significantly increased serum leptin levels, whereas the addition of the VV mixture significantly reduced serum leptin levels and markedly increased the adiponectin/leptin ratio (Figure 3H–J; C vs. H, H vs. HV groups). A 10% sticky Japanese diet mixture significantly reduced leptin mRNA expression in adipose tissues, consistent with findings that viscous polysaccharide intake lowers circulating leptin levels and adiposity in rodent models [20]. Leptin is an adipokine primarily secreted by adipose tissues and is closely associated with the fat mass and inflammation. The observed elevation in serum leptin levels in the HF group is consistent with previous findings that link HF diet-induced obesity to hyperleptinemia and leptin resistance [48]. Interestingly, the addition of the VV mixture significantly reduced serum leptin concentrations, suggesting an improvement in leptin sensitivity or a reduction in adipose tissue inflammation. Furthermore, the adiponectin/leptin ratio, a recognized biomarker reflecting the balance between anti-inflammatory and proinflammatory adipokines, was markedly higher in the HV group [49]. The diet used in this study was formulated based on the Japanese sticky vegetable mixture described by Hirokawa et al. (2019) [20]. Similar to their findings, a reduction in plasma leptin levels was observed in the present study, although Hirokawa et al. measured leptin mRNA in adipose tissue. This suggests that high-fiber vegetable mixtures may exert anti-leptin resistance effects and contribute to improved metabolic profiles.

The lipid metabolism-related gene expression analysis revealed that the HF diet significantly downregulated mRNA levels of key fatty acid oxidation enzymes, including *CPT1* and *ACO1*, whereas supplementation with the VV mixture effectively reversed these reductions (Figure 4L,M; C vs. H, H vs. HV groups). HF-diet-induced obesity is often associated with impaired β-oxidation due to downregulation of enzymes involved in fatty acid metabolism [50]. CPT1 serves as the rate-limiting enzyme for mitochondrial β-oxidation, while ACO1 initiates peroxisomal β-oxidation of very long-chain fatty acids [51]. Therefore, the suppressed expression of CPT1 and ACO1 in the HF group indicates reduced mitochondrial and peroxisomal fatty acid oxidation, contributing to hepatic lipid accumulation and steatosis. The upregulation of these genes by the VV mixture suggests enhanced hepatic lipid utilization. These effects may be attributed to viscous dietary fibers and polyphenols in the VV mixture, which were reported to activate AMPK signaling, a critical regulator of lipid oxidation [33,52]. Although many studies demonstrated that purified polysaccharides or polyphenol-rich extracts inhibit lipogenesis, such effects were not observed in this study. For example, okra-derived interventions were shown to suppress hepatic lipogenic genes, including lysophosphatidic acid acyltransferase (*LPAAT*), *lipin-1*, diacylglycerol O-acyltransferase 1 (*DGAT1*), and SREBP1 [18,31]. Additionally, crude polysaccharides from white tremella (*Tremella fuciformis*) at 200 mg/kg BW were reported to downregulate ACC, a rate-limiting enzyme in fatty acid synthesis, further supporting lipid-lowering effects [39]. Studies also found that 4% *Flammulina velutipes* polysaccharide supplementation upregulated mRNA expressions of *CD36*, *CPT1α*, and *MCAD*, while downregulating fatty acid synthase (*FAS*) expression [53]. However, the relatively short 8-week intervention in this study may have been insufficient to observe significant inhibition of fatty acid synthesis.

It should be noted that while lipid metabolism-related mRNA levels were measured, protein-level validation was not performed in this study. Previous research has shown that mRNA expression does not always correlate directly with protein abundance due to various regulatory mechanisms at post-transcriptional and translational levels [54,55]. Therefore, future studies incorporating protein expression analyses are necessary to better understand the functional implications of these gene expression changes.

### 3.4. Regulation of the Fecal Microbiotic Composition by a Homogenized VV Mixture

In the H group, the lengths of the small intestine and colon were found to have decreased (Figure 5A,B). According to Soares et al. (2015) [56], an HF diet promotes a 10% reduction in small intestine length. This effect is thought to result from an increased density of inhibitory nitrergic neurons, which reduce intestinal motility and increase food retention. These neuronal changes may reflect morphometric alterations in the small intestine [56].

An HF diet also significantly reduced the α-diversity and showed a distinct fecal microbiotic composition, which indicates a loss of gut microbial richness and evenness (Figure 5). Reduced microbial diversity and gut dysbiosis have been widely linked to metabolic disorders, including obesity and NAFLD, through mechanisms involving systemic inflammation and impaired lipid metabolism [57]. In the fecal microbiota analysis, the dominant bacterial taxa in the H group included the Bacteroidaceae (family), Bacteroides (genus), and Tannerellaceae (family) of the *Bacteroidota* phylum, as well as the Ruminococcaceae (family), Roseburia (genus), and Tuzzerella (genus) of the *Firmicutes* phylum (Figure 6). This microbial pattern is commonly associated with gut dysbiosis induced by HF diets [32]. Previous studies reported that increases in *Firmicutes*, particularly members of the Oscillospiraceae and Ruminococcaceae families, may promote energy harvesting and improve energy efficiency, thereby contributing to the development of obesity and metabolic disorders [58,59,60]. Ruminococcaceae and Roseburia are also recognized for their capacity to produce short-chain fatty acids (SCFAs), such as butyrate, which are known to modulate host energy metabolism and inflammation.

In the present study, supplementation with the VV mixture significantly increased the relative length of the small intestine (Figure 5A), enhanced fecal microbiota diversity and richness, and resulted in a microbial composition more similar to that of the C group (Figure 5F,G). After VV mixture supplementation, *Firmicutes* became the dominant phylum, particularly members of the class *Clostridia* and the families Oscillospiraceae and Lachnospiraceae, including the *Lachnospiraceae_NK4A136_*group (genus). These taxa are known for fermenting non-starch polysaccharides into SCFAs, especially butyrate, which plays key roles in maintaining gut barrier integrity, regulating immune responses, and modulating lipid metabolism [61,62,63].

Notably, the *Lachnospiraceae_NK4A136_*group has been associated with anti-inflammatory effects and improved lipid profiles in several dietary intervention studies [32,64]. The enrichment of this and other SCFA-producing bacteria suggests that the VV mixture may exert metabolic benefits, at least in part, through microbial modulation. Previous studies have also shown that dietary supplementation with viscous or fiber-rich components, such as okra powder, Tremella polysaccharides, purple yam resistant starch, and *Laminaria japonica*, promotes the growth of SCFA-producing taxa, including Lachnospiraceae and Muribaculaceae [22,23,65]. Similarly, the abundance of Muribaculaceae, another key SCFA-producing family, has been shown to increase following interventions with inulin and *Grifola frondosa* (maitake mushroom) [66,67]. On the other hand, Recent evidence indicates that *Firmicutes* are highly responsive to dietary fiber and play important roles in gut and metabolic homeostasis [68]. On the other hand, other studies have shown that specific *Firmicutes* taxa, such as *Lactococcus*, *Streptococcaceae*, and *Enterococcaceae*, are enriched in individuals with obesity or following a high-fat diet [69]. These seemingly contradictory observations highlight the need to interpret *Firmicutes* abundance in the context of specific taxa and dietary background.

Collectively, the observed shifts in microbial composition suggest that the VV mixture may exert its metabolic benefits, particularly the attenuation of HF diet-induced obesity and hepatic steatosis, through the enrichment of SCFA-producing bacteria and restoration of a healthier gut microbiota profile. However, the current study lacks direct measurement of fecal SCFA concentrations and functional gene profiling. Functional metagenomic analysis and SCFA quantification are planned in future studies to further elucidate the mechanistic links between microbial changes and host metabolic outcomes [59,60].

### 3.5. Research Applications and Limitations

These findings highlight the potential of using whole-food mucilaginous vegetable blends in developing texture-modified dietary interventions for older adults. Unlike purified extracts, such formulations are more closely aligned with habitual dietary practices and may offer additional nutritional and functional benefits through the synergistic effects of diverse dietary fiber and bioactive compounds.

However, this study has several limitations. First, the formulation ratio of the VV mixture was fixed and not optimized for maximum efficacy. Second, regarding viscosity, future studies should consider applying more-comprehensive rheological measurements using mechanical viscometers to obtain quantitative data. Third, the hepatic TBARS level was slightly elevated in the CV group, long term effects on oxidative stress of VV mixture supplementation should be assessed in future study. Fourth, SCFAs, along with targeted and non-targeted lipid metabolomics, which are highly relevant to the gut-liver axis and lipid metabolism [70], were not included in this study. Incorporating these analyses in future work will help provide a more comprehensive understanding of microbiota-mediated mechanisms. Finally, due to the complex composition of the vegetable blend, it was difficult to attribute the observed effects to any single component.

## 4. Materials and Methods

### 4.1. Preparation of a Homogenized Viscous Vegetable (VV) Mixture

According to Hirokawa et al., 2019 [20], Taiwanese local VVs were selected as substitutes, including seaweed (Laminariaceae), kelp (*Undaria pinnatifida*), agar (Gelidiaceae), white tremella (*Tremella fuciformis*), shiitake mushroom (*Lentinula edodes*), yellow strain *Flammulina velutipes*, okra (*Abelmoschus esculentus*), laver (root of *Pyropia*), purple yam (*Dioscorea alata*), and brown shimeji mushroom (*Hypsizygus tessellatus*). Vegetables were purchased from a local market. After rapid vacuum freeze-drying, each vegetable was ground into a powder and combined at equal weights (10 g per ingredient), such that each component accounted for 10% of the total mixture by weight. The nutrients and dietary fiber content were analyzed by SGS Taiwan (New Taipei City, Taiwan). Dietary fiber was analyzed using the AOAC 991.43 method [71,72]. Total polysaccharide and polyphenol contents in ten ingredients of VV were analyzed by MedCare Biotech (New Taipei City, Taiwan). The total polyphenol content is expressed as gallic acid equivalents (GAE) (mg GAE/g).

### 4.2. Viscosity Analysis

The VV mixture was dissolved in boiling water at a fixed ratio, then cooled to 37 °C. Viscosity was assessed using the Line Spread Test (LST) grading method and the International Dysphagia Diet Standardisation Initiative (IDDSI) framework [73,74].

### 4.3. Animals and Diets

Twenty-eight 6-week-old male C57BL/6 mice were purchased from BioLasco Taiwan (Ilan, Taiwan). All mice were housed in an animal room with a 12 h light/dark cycle at 22 ± 2 °C with 50–70% humidity. After 1 week of acclimation, mice were assigned to the following four groups based on body weight (BW): control diet (C), control diet with 10% VV mixture (CV), HF diet (H) and HF diet with 10% VV mixture (HV). Diet compositions are presented in Table 5. Each mouse was fed 3500 mg of diet per day, either without (C and H groups) or with 10% (i.e., 350 mg) of the diet replaced by the VV mixture (CV and HV groups). After 8 weeks, all mice were sacrificed, and blood samples were collected in tubes without anticoagulants for further analysis. Liver and adipose tissues were preserved at −80 °C for subsequent analyses. The Institutional Animal Care and Use Committee of Taipei Medical University approved all study procedures (LAC2024-0185).

### 4.4. Obesity-Related Indicators

Indicators of anti-obesity effects included BW changes and relative adipose tissue masses (perirenal, mesentery, and epididymal). The food efficiency ratio (FER) was calculated by the equation: FER = (body weight gain (g)/food intake (g)). The size of epididymal adipocytes was examined and semi-quantified using Image-Pro Plus 4.5 software (Media Cybernetics, Rockville, MD, USA). Adipose tissues were fixed in 10% buffered formalin, dehydrated in absolute ethanol overnight, embedded in paraffin, and sectioned at a thickness of 4 μm. Sections were then stained with hematoxylin and eosin (H&E) and evaluated by a pathologist using a light microscope (Olympus BX-51, Tokyo, Japan) equipped with a CCD camera [75].

### 4.5. Liver Damage

#### 4.5.1. Liver Function Index

Serum aspartate aminotransferase (AST) and alanine aminotransferase (ALT) activities were analyzed with the ADVIA^®^ Chemistry XPT System (Siemens Healthcare Diagnostics, Eschborn, Germany).

#### 4.5.2. Histopathological Examinations

Tissue section preparation, staining, and pathological interpretation were commissioned to Yu-An Biotechnology (Kaohsiung, Taiwan). Liver tissues were fixed in 10% buffered formaldehyde, and then sections were stained with Harris’ hematoxylin and eosin (H&E). Images were interpreted by pathologists under an optical microscope (Olympus BX-51, Tokyo, Japan) equipped with a CCD camera. Histopathological evaluations of macrovesicular steatosis, microvesicular steatosis, hypertrophy, and the number of inflammatory foci were separately scored. The NAFLD score was the sum of the above four parameters [76].

#### 4.5.3. Lipid Peroxidation Content

Lipid peroxidation is an indicator of oxidative stress, as measured by a thiobarbituric acid-reactive substance (TBARS) kit (TBARS 10009055 Assay Kit, Cayman Chemical, Ann Arbor, MI, USA). The lipid peroxidation content was presented as malondialdehyde (MDA) levels.

### 4.6. Lipid Metabolism-Related Factors

#### 4.6.1. Serum Lipid Profiles

Total cholesterol (TC), total triglycerides (TGs), high-density lipoprotein cholesterol (HDL-C), and low-density lipoprotein cholesterol (LDL-C) were detected with the ADVIA^®^ Chemistry XPT System (Siemens Healthcare Diagnostics).

#### 4.6.2. Hepatic TC and TG Concentrations

Hepatic lipids were extracted according to the method from the kit instructions. Hepatic TC and TG concentrations were analyzed with a cholesterol colorimetric assay kit (STA-390, Cell Biolabs Inc., San Diego, CA, USA) and colorimetric TG assay kit (Triglyceride assay kit no.10010303, Cayman Chemical, Ann Arbor, MI, USA), and results are expressed as milligrams per gram (mg/g) of liver tissue.

#### 4.6.3. Plasma Adipokines Levels

Plasma adipokines levels were detected with a Leptin Mouse/Rat ELISA kit (Biovendor, Brno, Czech Republic) and a Rat Adiponectin ELISA kit (Assaypro, Charles, MO, USA), following the manufacturer’s instructions.

#### 4.6.4. Hepatic Fatty Acid and Cholesterol Metabolism-Related Gene Messenger (m)RNA Levels

A real-time polymerase chain reaction (PCR) was used to evaluate hepatic lipid metabolism-related gene mRNA levels. Total RNA of the liver was extracted with the TRI Reagent^®^ (Sigma-Aldrich, St. Louis, MO, USA). The quality and quantity of total RNA were evaluated by measuring the optical density (OD) 260/280 nm ratio on a BioTek epoch reader with the Gen5TM Take3 Module (BioTek Instruments, Winooski, VT, USA). Total RNA (4000 ng/µL) was reverse-transcribed with a RevertAid First Strand cDNA Synthesis kit (#K1621, ThermoFisher Scientific, Waltham, MA, USA). The concentration of complementary (c)DNA was calculated by the BioTek epoch reader with the Gen5TM Take3 Module system and adjusted to 50 ng/µL. After that, cDNA was amplified in a 96-well PCR plate using SYBR Green/ROX qPCR Master Mix (2×) (ThermoFisher Scientific, Waltham, MA, USA) on a QuantStudio 1 Real-Time PCR System (ThermoFisher Scientific, Waltham, MA, USA). β-actin was used as internal control, and mRNA levels were calculated by 2^−ΔΔCT^. Gene levels were normalized to β-actin, and the ratio to β-actin was calculated by setting the value of the NC group to 1. Information on primers is given in Table 6.

### 4.7. Intestinal Damage

#### 4.7.1. Relative Intestine Length

The small intestine and colon of each animal were measured and presented as the relative length in reference to the BW, the formula was as follow, intestine or colon length/body weight. The small intestine was collected from the end of pylorus to ileocecal valve, while the colon was collected from the end cecum to the beginning of anus.

#### 4.7.2. Fecal Microbiotic Analysis

A fecal microbiotic analysis and interpretation were performed by the Taipei Medical University Joint Biobank (Taipei, Taiwan). Fresh fecal samples were collected and stored at −80 °C until analysis. 16S ribosomal (r)RNA extraction and purification were conducted using the QIAamp^®^ Fast DNA Stool Mini Kit (Qiagen, Hilden, Germany). Sequencing of the V3–V4 regions was performed using the Illumina MiSeq platform. A microbial community analysis was conducted using the phyloseq package (v1.30.0). The Chao1 index was used to assess alpha diversity, while a principal coordinates analysis (PCoA) based on variance-adjusted weighted UniFrac distances was applied to evaluate the beta diversity. The linear discriminant analysis effect size (LEfSe) was used to identify differentially abundant taxa among the four groups.

### 4.8. Statistical Analysis

Data from the various parameters measured are expressed as the mean ± standard deviation (SD). Statistical analyses were performed with GraphPad Prism 9.0 software (La Jolla, CA, USA). Student’s *t*-test was used to determine statistical differences between groups in pairwise comparisons, such as C vs. CV, C vs. H, CV vs. HV, and H vs. HV. A *p* value of <0.05 was considered statistically significant.

## 5. Conclusions

This study demonstrated that supplementation with the mucilaginous vegetable (VV) mixture effectively reduced body weight, adipose tissue mass, and adipocyte size, thereby exhibiting significant anti-obesity effects in mice fed a high-fat diet. Additionally, the VV mixture alleviated liver damage and lipid metabolic disturbances by promoting fatty acid oxidation. It also modulated the gut microbiotic composition by increasing short-chain fatty acid-producing bacteria, such as *Lachnospiraceae_NK4A136_group*, which supports intestinal health. These findings suggest that mucilage-rich food blends may have potential for preventing high-fat diet-induced obesity possibly through regulating hepatic fatty acid oxidation and the gut microbiotic composition.

## Figures and Tables

**Figure 1 plants-14-02510-f001:**
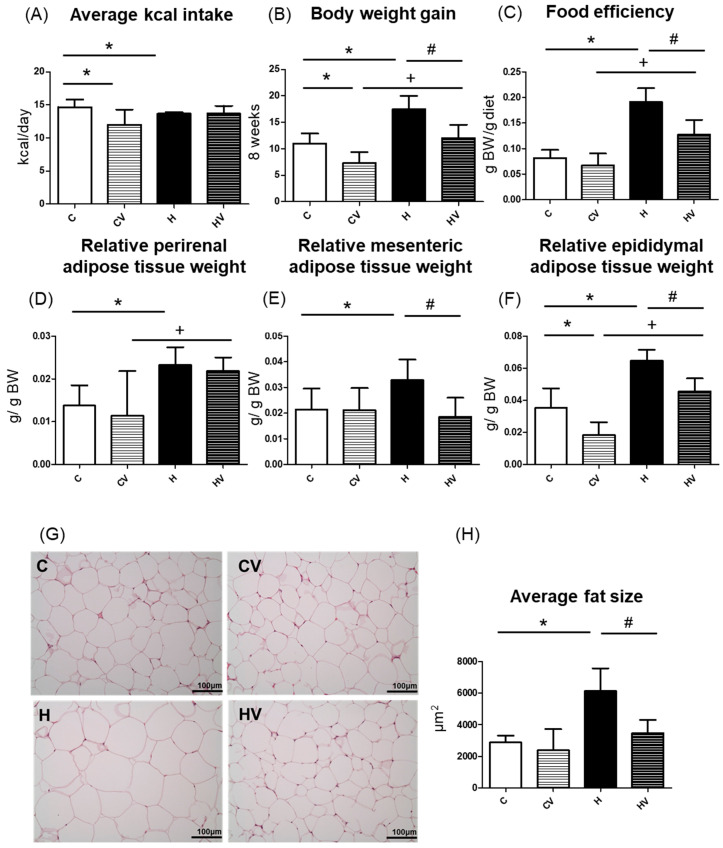
Effects of the homogenized viscous vegetable mixture on food intake and body weight in high-fat (HF) diet-fed mice. (**A**) Body weight gain, (**B**) average calorie intake, (**C**) food efficiency ratio (FER), (**D**) relative perirenal adipose tissue weight, (**E**) relative mesenteric adipose tissue weight, (**F**) relative epididymal white adipose tissue weight, (**G**) H&E staining results of epididymal adipose tissue (magnification: 200×; scale bar: 100 μm) (*n* = 5), (**H**) adipose cell average size (*n* = 5). The FER was calculated by applying the equation: FER = (body weight gain (g)/food intake (g)). Values are presented as the mean ± SD (*n* = 5–7). Significance between two groups was determined using Student’s *t*-test. *p* < 0.05. * represents a significant difference with the C group, ^#^ represents a significant difference with the H group; ^+^ represents a significant difference with the CV group.

**Figure 2 plants-14-02510-f002:**
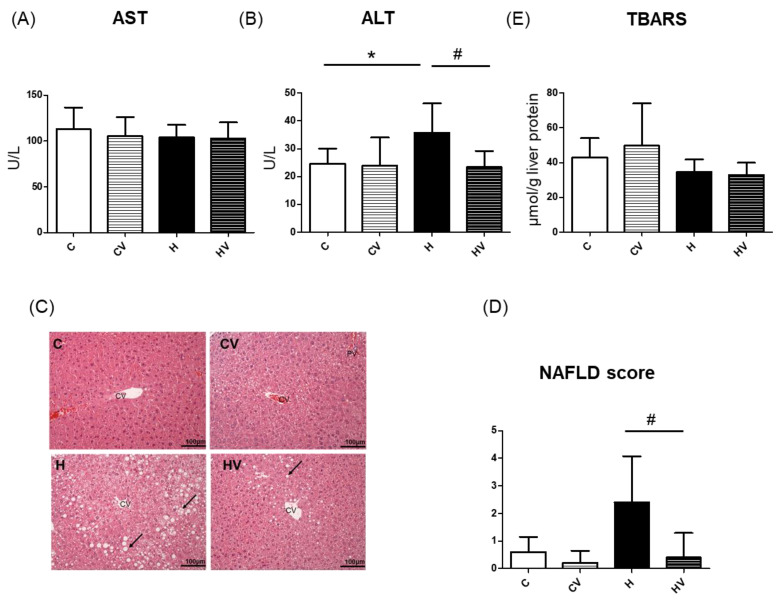
Effects of the homogenized viscous vegetable mixture on liver damage in high-fat (HF)-diet-fed mice. (**A**) Serum aspartate aminotransferase (AST), (**B**) serum alanine aminotransferase (ALT), (**C**) H&E staining results of liver tissue sections (magnification: 200×; scale bar: 100 μm) (*n* = 5), (**D**) nonalcoholic fatty liver disease (NAFLD) score (*n* = 5), (**E**) lipid peroxidation indicator, thiobarbituric acid-reactive substances (TBARSs). Data are expressed as the mean ± standard deviation (SD) (*n* = 5–7). Significance between two groups was determined using Student’s *t*-test. *p* < 0.05. * represents a significant difference with the C group, ^#^ represents a significant difference with the H group. The black arrow indicates hepatic steatosis. CV, central vein.

**Figure 3 plants-14-02510-f003:**
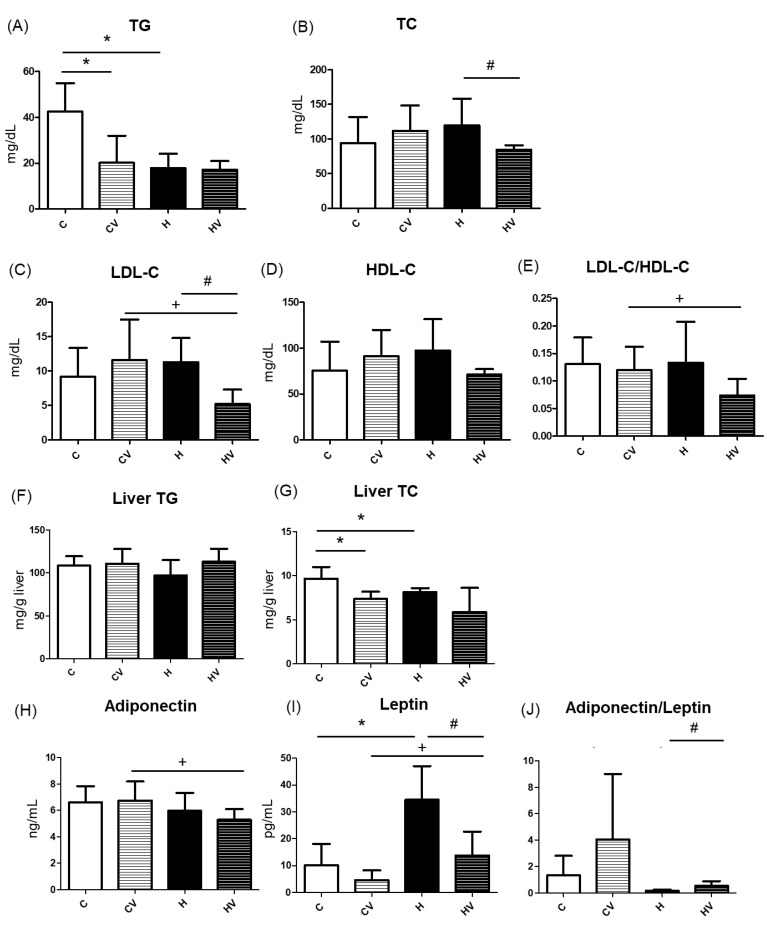
Effects of the homogenized viscous vegetable mixture on lipid metabolism in high-fat (HF) diet fed mice. (**A**) Serum triglyceride (TG), (**B**) serum total cholesterol (TC), (**C**) serum low-density lipoprotein cholesterol (LDL-C), (**D**) serum high-density lipoprotein cholesterol (HDL-C), (**E**) LDL-C/HDL-C ratio, (**F**) liver TG, (**G**) liver TC, (**H**) plasma adiponectin, (**I**) plasma leptin, (**J**) adiponectin/leptin ratio. Data are expressed as the mean ± standard deviation (SD) (*n* = 7). Significance between two groups was determined using Student’s *t*-test. *p* < 0.05. * represents a significant difference with the C group, ^#^ represents a significant difference with the H group; ^+^ represents a significant difference with the CV group.

**Figure 4 plants-14-02510-f004:**
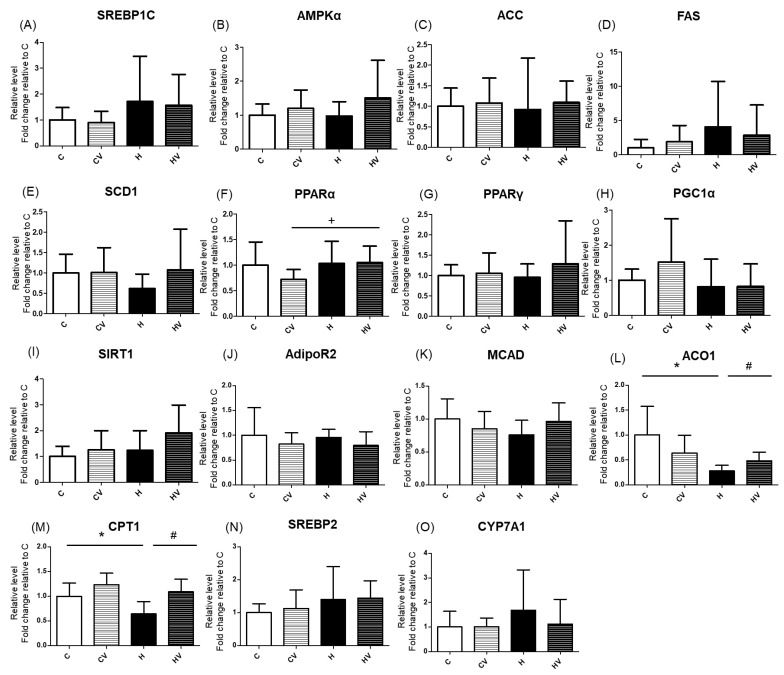
Effects of the homogenized viscous vegetable mixture on hepatic mRNA expressions related to fatty acid synthesis (**A**–**E**), fatty acid oxidation (**F**–**M**) and cholesterol-metabolism (**N**,**O**) in high-fat (HF) diet-fed mice. Data are expressed as the mean ± standard deviation (SD) (*n* = 7). Significance between two groups was determined using Student’s *t*-test. *p* < 0.05. * represents a significant difference with the C group, ^#^ represents a significant difference with the H group; ^+^ represents a significant difference with the CV group. SREBP, sterol regulatory element-binding protein; AMPKα, adenosine monophosphate-activated protein kinaseα; ACC, acetyl-CoA carboxylase; FAS, fatty acid synthase; SCD1, stearoyl-CoA desaturase 1; PPAR, peroxisome proliferator-activated receptor; PGC1, peroxisome proliferator-activated receptor gamma coactivator 1; SIRT1, sirtuin 1; AdipoR2, adiponectin receptor 2; MCAD, medium-chain acyl-CoA dehydrogenase; ACO1, acyl-CoA oxidase 1; CPT1, carnitine palmitoyltransferase 1; CYP7A1, cholesterol 7 alpha-hydroxylase.

**Figure 5 plants-14-02510-f005:**
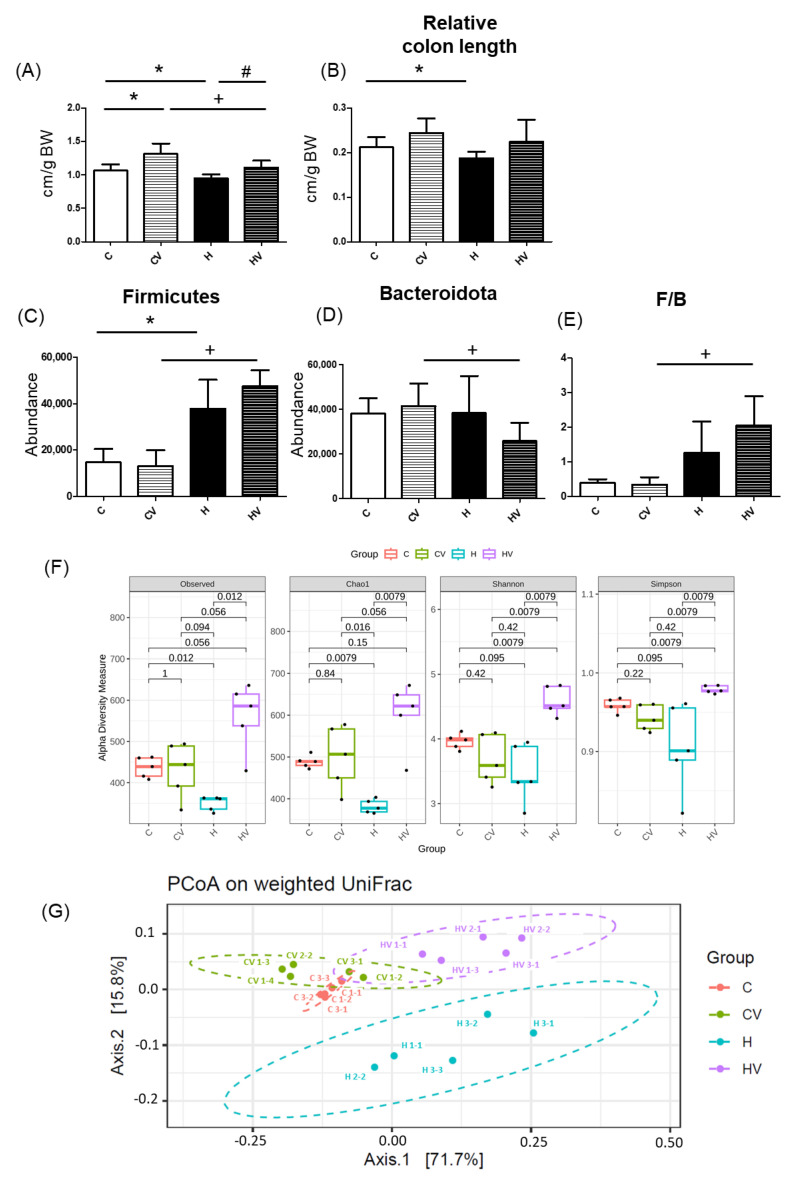
Effects of the homogenized viscous vegetable mixture on (**A**) the relative small intestine length (*n* = 7), (**B**) relative colon length (*n* = 7), (**C**) Firmicutes abundance, (**D**) Bacteroidota abundance, (**E**) Firmicutes/Bacteroidota ratio, (**F**) α-diversity and (**G**) a principal coordinate analysis (PCoA) of the fecal microbiota in mice with high-fat (HF) diet feeding. Data are expressed as the mean ± standard deviation (SD) (*n* = 5–7). Significance between two groups was determined using Student’s *t*-test. *p* < 0.05. * represents a significant difference with the C group, ^#^ represents a significant difference with the H group; ^+^ represents a significant difference with the CV group.

**Figure 6 plants-14-02510-f006:**
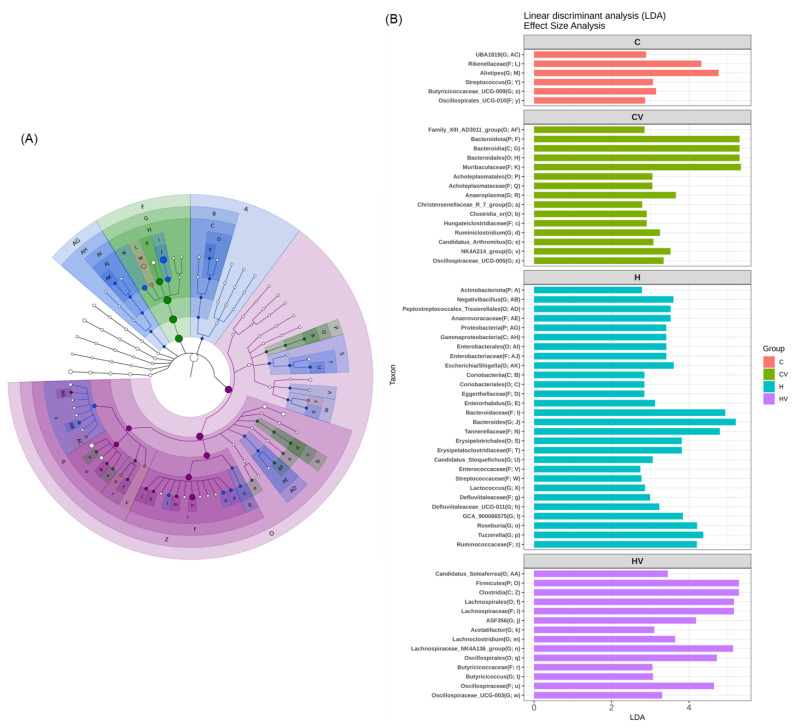
Effects of the homogenized viscous vegetable mixture on taxonomies of fecal microbiotic compositions in mice with high-fat-diet feeding. (**A**) A linear discriminant analysis of the effect size (LEfSe) of the most significant abundance differences in the fecal microbiota among all groups (*n* = 5). (**B**) Bacteria meeting the linear discriminant analysis (LDA) threshold (≥2) differed among all groups (*n* = 5).

**Table 1 plants-14-02510-t001:** Components of the homogenized viscous vegetable mixture.

	Viscous Vegetable Mixture (100 g)
Crude protein (g)	17.7
Crude fat (g)	3.8
Saturated fat (g)	1.25
Trans fat (g)	-
Carbohydrates (g)	62.5
Soluble dietary fiber (g)	16.0
Insoluble dietary fiber (g)	26.1
Sugar (g)	2.7
Ash (g)	7.5
Sodium (mg)	773.7
Moisture (g)	8.5
Calories (kcal)	270.8

**Table 2 plants-14-02510-t002:** Total polysaccharide and polyphenol contents of the homogenized viscous vegetable.

	Total Polysaccharides (mg/g)	Total Polyphenols (mg GAE/g)
Seaweed (*Laminariaceae*)	13.7	1.2
Kelp (*Undaria pinnatifida*)	4.5	0.4
Agar (Gelidiaceae)	1.1	0.03
White tremella (*Tremella fuciformis*)	1.7	0.08
Shiitake mushroom (*Lentinula edodes*)	18.3	1.2
Yellow strain *Flammulina velutipes*	25.6	1.8
Okra (*Abelmoschus esculentus*)	122.7	3.4
Laver (root of *Pyropia*)	2.6	2.2
Purple yam (*Dioscorea alata*)	7.9	0.4
Brown shimeji mushroom (*Hypsizygus tessellatus*)	13.9	1.3
Total	212	12

GAE, gallic acid equivalents.

**Table 3 plants-14-02510-t003:** Viscosity classification of the homogenized viscous vegetable mixture using the Line Spread Test (LST) ^1^.

Ratio ^2^	Quadrant 1	Quadrant 2	Quadrant 3	Quadrant 4	Quadrant 5	Quadrant 6	LST Value	Classification ^3^
1:9	35	36.5	32.5	30.5	33.5	31	33.2	Grade 2
1:12	44	43	47.5	49.5	51	43.5	46.4	Below threshold
1:14	42	41.5	44	51	53.5	49	46.8	Below threshold
1:17	45.5	47	44.5	50	50	47	47.3	Below threshold
1:19	52	51	52.5	56	58	57.5	54.5	Below threshold

^1^ The homogenized viscous vegetable mixture was dissolved in boiling water, stirred, and cooled to 37 °C at room temperature before the measurement. ^2^ Ratio of mucilaginous vegetable powder (g) to hot water (mL). ^3^ Grade 2 indicates “moderately thick”.

**Table 4 plants-14-02510-t004:** Viscosity classification of the homogenized viscous vegetable mixture using the IDDSI framework ^1^.

Ratio ^2^	Volume Remaining in Syringe After 10 s (mL)	Classification ^3^
1:9	10	Level 4
1:12	9	Level 3
1:14	9	Level 3
1:17	8	Level 3
1:19	8	Level 3

^1^ The homogenized viscous vegetable mixture was dissolved in boiling water, stirred, and cooled to 37 °C at room temperature before the measurement. ^2^ Ratio of mucilaginous vegetable powder (g) to hot water (mL). ^3^ Level 4 indicates “extremely thick,” and level 3 indicates “moderately thick.

**Table 5 plants-14-02510-t005:** Compositions of the experimental diets.

	C	CV	H	HV
Protein (kcal%)	14.7%	15.5%	19.9%	20.2%
Carbohydrates (kcal%)	75.6%	72.3%	19.3%	19.9%
Fat (kcal%)	9.5%	9.7%	60.5%	57.9%
kcal/g	3.81	3.70	5.25	5.00
Ingredients (g/kg)				
Cornstarch ^1^	465	418.5	0	0
Maltodextrin ^2^	155	139.5	163.4	147.06
Sucrose ^3^	100	90	90	81
Casein ^4^	140	126	261.5	235.35
L-cysteine ^5^	2	1.8	3.9	3.51
Soybean oil ^6^	40	36	32.7	29.43
Lard ^7^	0	0	320.4	288.36
Cellulose ^8^	50	45	65.4	58.86
Mineral mixture (AIN-93M-MIX) ^9^	35	31.5	45.8	41.22
Vitamin mixture (AIN-93M-MIX) ^10^	10	9	13.1	11.79
Choline bitarate ^11^	3	2.7	3.9	3.51
Tert-butylhydroquinone ^12^	0.008	0.0072	0.01	0.009
Homogenized viscous vegetable mixture ^13^	0	100	0	100

C, normal control group, AIN-93M (10% fat); CV, C + 10% homogenized viscous vegetable mixture; H, high-fat diet group, AIM-93M (60% fat); HV, H + 10% homogenized viscous vegetable mixture. ^1^ Cornstarch: 902956, MP Biomedicals, Irvine, CA, USA. ^2^ Maltodextrin: 960429, MP Biomedicals. ^3^ Sucrose: Taiwan Sugar Corporation, Taipei, Taiwan. ^4^ Casein: 901293, MP Biomedicals. ^5^ L-cysteine: 101454, MP Biomedicals. ^6^ Soybean oil: Taiwan Sugar Corporation. ^7^ Lard: 902140, MP Biomedicals. ^8^ Cellulose: 900453, MP Biomedicals. ^9^ Mineral mixture (AIN-93M-MIX): 960401, MP Biomedicals. ^10^ Vitamin mixture (AIN-93M-MIX): 2960402, MP Biomedicals. ^11^ Choline bitarate: 101384, MP Biomedicals. ^12^ Tert-butylhydroquinone: 195590, MP Biomedicals. ^13^ Homogenized viscous vegetable mixture: a mixture of ten vegetables, the components are shown in Section 4. 100 g contains crude protein 17.7 g, crude fat 3.8 g, carbohydrates 62.5 g, soluble dietary fiber 16 g, insoluble dietary fiber 26.1 g, and 270.8 kcal.

**Table 6 plants-14-02510-t006:** Primers used for the quantitative polymerase chain reaction.

	Forward 5′ → 3′	Reverse 5′ → 3′
SREBP1c	AGATCCAGGTTTAGGTGGG	ATCGCAAACAAGCTGACCTG
AMPKα	TGATGTGAGGGTGCCTGAAC	GAAAGTGAAGGTGGGCAAGC
ACC1	GGACCACTGCATGGAATGTTAA	TGAGTGACTGCCGAAACATCTC
FAS	AACCTGATGGATGAGCACC	CTGTGCCCGTCGTCTATACC
SCD1	CCTCCTGCAAGCTCTACACC	CTGCCTTGGGTCAGAGGGTA
PPARα	TTGCAGCTTCGATCACACTTGTCG	TACCACTATGGAGTCCACGCATGT
PPARγ	ACCTGATGGCATTGTGAGACA	ATTGAGTGCCGAGTCTGTGG
PGC1	GGAATATGGTGATCGGGAACA	AAAGGATGCGCTCTCGTTCA
SIRT1	TTGACCGATGGACTCCTCACT	ATTGTTCGAGGATCGGTGCC
AdipoR2	AGAATCCGTGGAGCTCAGCA	TGTCCAAATGTTGCCCGTCT
MCAD	AACTAAACATGGGCCAGCGA	GAAACCTGCTCCTTCACCGA
ACO1	TTTGTGGAACCTGTTGGCCT	AAAATCTGGGGCTCTGGCTC
CPT1	ACTCCGCTCGCTCATTCCG	GAGATCGATGCCATCAGGGG
SREBP2	TGAGTACATGTGGGGAGCTT	TCAAACCCCACGGCAACAA
CYP7A1	GGGCAGGCTTGGGAATTTTG	ACAGCTACTAGGGGGCTTCA
*β* actin	CTGAGCTGCGTTTTACACCC	TTTGGGGGATGTTTGCTCCA

SREBP1c, sterol regulatory element-binding protein 1c; AMPKα, adenosine monophosphate-activated protein kinaseα; ACC, acetyl-CoA carboxylase; FAS, fatty acid synthase; SCD1, stearoyl-CoA desaturase 1; PPARα, peroxisome proliferator-activated receptor α; PPARγ, peroxisome proliferator-activated receptor γ; PGC1, peroxisome proliferator-activated receptor gamma coactivator 1; SIRT1, sirtuin 1; AdipoR2, adiponectin receptor 2; MCAD, medium-chain acyl-CoA dehydrogenase; ACO1, acyl-CoA oxidase 1; CPT1, carnitine; SREBP2, sterol regulatory element-binding protein 2; CYP7A1, cholesterol 7 alpha-hydroxylase.

## Data Availability

The data that support the findings of this study are available from the corresponding author upon reasonable request.

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
