# Peer review of "Gut–Liver Axis-Mediated Anti-Obesity Effects and Viscosity Characterization of a Homogenized Viscous Vegetable Mixture in Mice Fed a High-Fat Diet"

_plants, 2025, doi:10.3390/plants14162510_

Round 1
Reviewer 1 Report
Comments and Suggestions for Authors
The manuscript provides experimental data and discussion on the Gut-liver axis-mediated anti-obesity effects and viscosity characterization of a homogenized viscous vegetable mixture in mice fed a high-fat diet. The overall completeness of the experimental work is excellent, but there are also some issues and shortcomings. The following are detailed comments:
- In section 2.3. Lipid metabolism-related factors, the authors discuss the lipid metabolism of each group of mice in the experiment. The measurement of certain parameters, such as hepatic TC and TG concentrations, is reasonable. However, it should be emphasized that the measurement and provision of ten recognized indicators that reflect lipid metabolism can explain the problem, and the measurement and provision of one recognized indicator that reflects lipid metabolism can also explain the problem. The difference lies in the fact that more comprehensive data leads more convincing. Therefore, authors need to consider using non-targeted metabolomics or targeted lipid metabolomics to provide stronger and more comprehensive evidence of lipid metabolism in mice.
- Regarding section 2.3.4. Lipid metabolism-related mRNA levels, although it is generally believed that changes at the gene level reflect changes at the protein level, the process of mRNA translation into proteins is influenced and regulated by various factors that cannot be ignored. Therefore, the authors may need to provide protein-level validation to support their statements regarding lipid metabolism pathways.
- The author's discussion and analysis of gut microbes is incomplete. Not only does it neglect the analysis of functional genes in the gut microbiota, but it also fails to conduct any correlation analysis based on data indicators related to the function of the gut microbiota and lipid metabolism, liver, etc. Instead, it only provides descriptions and inferences based on existing research, which is not widely accepted.
- Although the author acknowledges the limitation of not measuring SCFAS in the study, given that the author's discussion of the gut microbiota includes a discussion of SCFAS-producing bacterial genera, it is inappropriate not to measure SCFAS.
Reviewer 2 Report
Comments and Suggestions for Authors
the manuscript by Yu-An Wei et al. presents a comprehensive and well-structured preclinical study evaluating the metabolic, hepatic, and gut microbiota-related effects of a homogenized mucilaginous vegetable (VV) mixture in mice subjected to a high-fat (HF) diet. The study addresses a relevant topic in the context of non-pharmacological dietary interventions for obesity and metabolic-associated fatty liver disease (MAFLD). A notable strength is the focus on whole-food blends, in contrast to isolated extracts, which better mimics real-world dietary applications, particularly for populations requiring texture-modified diets.
Overall, the study is scientifically sound, methodologically robust, and well-written. However, is the opinion of this Reviewer that only two points should be addressed to enhance scientific rigor and impact.
- SCFAs are central to the hypothesized mechanism of action via modulation of the gut-liver axis, yet they were not measured. Their absence limits mechanistic interpretation. The authors acknowledge this in the discussion, but a stronger emphasis on this limitation and a suggestion for follow-up studies is warranted.
- The HF group showed a paradoxical decrease in serum TGs, which the authors discuss reasonably. Still, the potential role of VLDL secretion impairment or altered lipoprotein lipase activity should be framed more cautiously, perhaps by referencing more relevant models or including suggestions for biochemical validation.
Reviewer 3 Report
Comments and Suggestions for Authors
Please see the attached file

The entire manuscript is well-written; however, it still needs some improvement. Please see the attached file
Author Response
Please referr to the attachment.
